

# An assessment of sub snow GPS for quantification of snow water equivalent

Ladina Steiner[1], Michael Meindl[1], Charles Fierz[2], and Alain Geiger[1]

[1]Institute of Geodesy and Photogrammetry, ETH Zurich, Robert-Gnehm-Weg 15, 8093 Zurich, Switzerland
[2]WSL Institute for Snow and Avalanche Research SLF, Flüelastr. 11, 7260 Davos, Switzerland

**Correspondence:** Ladina Steiner (ladinasteiner@ethz.ch)

**Abstract.** Global Navigation Satellite Systems (GNSS) contribute to various Earth observation applications. The present study investigates the potential and limitations of the Global Positioning System (GPS) to estimate in situ water equivalents of the snow cover (snow water equivalent, SWE) by using buried GPS antennas. GPS derived SWE is estimated over three seasons (2015/16 − 2017/18) at a high Alpine test site in Switzerland. Results are validated against state of the art reference sensors: snow scale, snow pillow, and manual observations. SWE is estimated with a high correspondence to the reference sensors for all three seasons. Results agree with a median relative bias below 10 % and are highly correlated to the mean of the three reference sensors. The sensitivity of the SWE quantification is assessed for different GPS ambiguity resolution techniques, as the results strongly depend on the GPS processing.

## 1   Introduction

Knowledge of snow cover characteristics is an important basis for climatology, natural hazards forecasting, early-warning systems, and hydro energy industries. Extensive amount of water stored in snow covers has a high impact on flood development during snow melting periods. High damage is caused worldwide by floods, originating from mountain catchment areas. Early assessment of the snow water equivalent (depth of water that would result if the mass of snow melted completely, Fierz et al., 2009) in mountain environments enhances early-warning and thus prevention of major flood events.

Several point wise measurement methods already exist to continuously determine snow water equivalent (SWE). SWE is measured usually in situ with manual or automated observation techniques and is expressed in units of mass per area ($kg/m^2$) or in millimetres of water equivalent (mm w.e.). Using SWE tubes, a sample is taken out of the snow profile and weighted afterwards leading to SWE. Furthermore, SWE is calculated indirectly based on snow depth and the bulk snow density, measured manually in a snow pit or along a transect (WMO, 2008; Sturm et al., 2010). Both techniques are state of the art and considered as most reliable at the moment. They are, however, labour-intense, time consuming, destructive, and have a low temporal resolution. Automated and continuous SWE measurements are provided by a snow pillow or a snow



scale (Beaumont, 1965, 1966; Johnson et al., 2007, 2015) and are, however, prone to errors especially during snow melt events. Additionally, cosmic ray neutron probes and other passive or acoustic instruments measure indirectly SWE (Harding, 1986; Kodama et al., 1979; Rasmussen et al., 2012; Kinar and Pomeroy, 2007, 2015a). Smith et al. (2017) and Kinar and Pomeroy (2015b) provide a detailed summary as well as a comprehensive description of terrestrial SWE measurement techniques. SWE

observations based on satellite remote sensing is limited to large plains due to low spatial resolution and problems with steep orography of mountain chains like the Alps. However, accurate and reliable in situ data is still needed for calibration and validation of remote sensing data (Goodison and Walker, 1995; Derksen et al., 2005; Takala et al., 1995).

Global navigation satellite system (GNSS) remote sensing techniques are capable to provide reliable, accurate, efficient, and continuous observations independent of weather conditions. Sub snow GNSS techniques are lately tested for snow water

equivalent estimation (Henkel et al., 2018), suggested to determine liquid water content (Koch et al., 2014) or considered for avalanche rescue (Claypool, 1997; Schleppe and Lachapelle, 2008; Olmedo et al., 2012). Most studies concentrate on the use of Global positioning system (GPS) single frequency signals (L1, with a frequency of $f_1 = 1575.42\,\mathrm{MHz}$). GPS antennas buried under snow were first used to evaluate the potential of GPS signal reception and positioning performance in avalanche rescue research (Claypool, 1997). Schleppe and Lachapelle (2008) analyzed experimentally the GPS tracking performance under

avalanche deposited snow at two test sites in Canada. The potential of a GPS based rescue system for victims buried under avalanches was investigated by Olmedo et al. (2012) in the framework of the SICRA project. Steiner et al. (2018) analyzed the GPS receiver behaviour from antennas submerged into water and developed a model to estimate the water depth above the submerged GPS antenna based on the path delay of the GPS L1 signals. Another study deals with the analysis of GPS L1 signal to noise ratio (SNR) for snow property estimation from a ground GPS antenna compared to a GPS reference antenna

above snow surface (Appel et al., 2014). Koch et al. (2014) estimated the liquid water content (wetness) continuously based on GPS L1 signal strength observations for a seasonal snowpack in the Swiss Alps. Henkel et al. (2018) show exemplary the possibility of SWE estimation using GPS L1 carrier phase residuals for the dry snow period during winter 2015/16.

Furthemore, GNSS stations above a snowpack are used for SWE estimation based on the interference of direct and reflected GNSS signals (GNSS reflectometry). Thereby, the SWE is estimated using GPS L1 C/A or L2c signal-to-noise ratio (e. g.

McCreight and Small, 2014; Jacobson, 2010, 2012). Different GNSS reflectometry methods are used for snowpack character-ization (e. g. Larson et al., 2009; Boniface et al., 2015; Ozeki and Heki, 2012; Najibi and Jin, 2013), mainly for snow depth estimation. Cardellach et al. (2012) investigated the potential to remotely sense sub surface snow structures in dry snow areas using bi-statically reflected GNSS signals.

This paper presents a method to estimate SWE based on phase based differential GPS by using an antenna buried in the

snowpack (sub snow GPS) and a reference station above the snowpack. A model is developed to estimate SWE based on the refraction and path delay of GPS code and phase signals while propagating through a snowpack. A sensitivity analysis on SWE estimation results is carried out analyzing different ambiguity resolution techniques within the differential GPS processing. Results are validated against the state of the art reference sensors snow pillow, snow scale, and manual observations for three full winter seasons at the high alpine test site Weissfluhjoch of the WSL Institute for Snow and Avalanche Research (SLF).



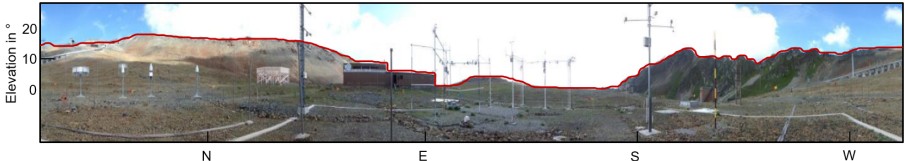

**Figure 1.** Horizon of the Weissfluhjoch test site (MeteoSwiss, 2014, modified).

Sect. 2 describes the study site, the geodetic GPS equipment, and the reference sensors. Sect. 3 derives the model to estimate the SWE from the GPS signal refraction and propagation delay. The SWE of a seasonal snowpack is estimated using buried GPS antennas and the method of investigation is described in Sect. 4. The developed model is applied to a seasonal snowpack over three winter seasons and results are analyzed in Sect. 5. Sect. 6 discusses the results and Sect. 7 concludes the paper.

## 2  Study site and instrumental set-up

The GPS snow monitoring system is installed in a snow free period at the SLF test site Weissfluhjoch. The network includes geodetic and low cost GPS stations, operating permanently since October 2012. The present study focuses on the geodetic station in order to understand the receiver behaviour. A follow up study will investigate the potential of the low cost system.

### 2.1  Weissfluhjoch test site

The Weissfluhjoch test site (Fig. 1) is located in the Swiss Alps above Davos at 2'536 m a.s.l. The 40 x 40 m flat test site is obstructed at elevations below 20 degrees by mountain faces in North (N) and West (W) direction and open regarding East (E) and South (S). The open E and S direction and sky visibility above 20 degrees in N and W enhances the GPS satellite visibility and thus the observation geometry. The main winter wind direction for the site comes from NW and SE (over the years 2001 – 2011, MeteoSwiss, 2014). The test site is equipped with energy supply, internet connection for automated data transmission, and reference data for validation of the GPS snow monitoring system (Fig. 2).

### 2.2  GPS snow monitoring system

The GPS snow monitoring system is set up at the Weissfluhjoch test site and consists of a geodetic GNSS system. However, as the present study only deals with GPS signals (for future comparison to low cost systems), the term GPS is used further on for all equipment. The use of a geodetic system allows a better understanding of all snow effects on the observations by minimizing additional effects. A GPS antenna (Leica AS10) is mounted at 5.3 m height on a pole and serves as reference station. A second GPS antenna (Leica AS10) is installed on the ground next to the reference station (Fig. 2) and is referred to as the sub snow GPS station. This set up results in a very short baseline for differential GPS processing. The corresponding GPS receivers (Leica GR10) are sheltered in the hut next to the test site, allowing power supply, internet connectivity for remote access as well as any time access in case of receiver problems. Dedicated low loss GPS cables (Huber+Suhner SPUMA 400) are used to



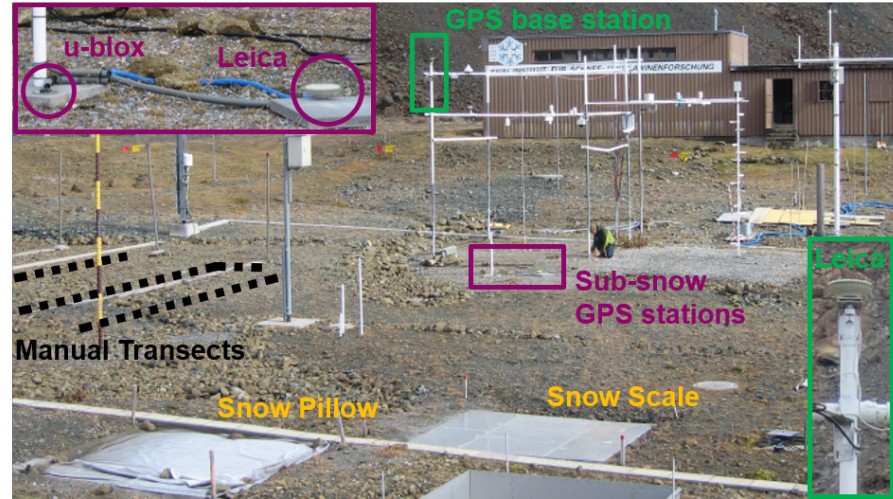

**Figure 2.** Weissfluhjoch test site, equipped with the GPS snow monitoring system, snow scale, and snow pillow. Manual transects are carried out bi-weekly at the North side as shown by the dashed line.

mitigate signal loss caused by the long path below the snow surface of approx. 40 m. The use of the same equipment for both stations (antenna, receiver, antenna cable type and length) ensures best possible consistency for the differential GPS processing.

In this set up, the sub snow GPS antenna is placed on the ground and is buried by snow after the first snowfall already. A GPS antenna is, however, designed for an usual environment of air. The antenna impedance matches the impedance of air in order to avoid refraction effects at the antenna/air boundary. When changing the environment by placing the antennas into snow, an antenna impedance mismatch could occur, influencing the tracking performance (Rao et al., 2011).

As the GPS system was installed in the snow free period, these observations have been used to determine best possible reference coordinates for the sub snow GPS station (Sect. 4). Later, SWE estimation results from the GPS monitoring system are validated against state of the art reference data operated by the SLF.

## 2.3   Reference data

SWE reference data (Marty, 2017) are provided by the SLF from the test site Weissfluhjoch (Fig. 2). A snow pillow (Sommer Messtechnik SP3) and a snow scale (Sommer Messtechnik SSG1000) serve as reference data for SWE estimation. The snow pillow measures the overlying pressure of the snowpack on a fluid filled bladder. The snow scale uses a weighing surface and load cells to measure the weight of the overlying snow (Beaumont, 1966; Johnson et al., 2007). The sensors are located on the ground, 20 m next to the sub snow GPS station (Fig. 2). Both sensors are offset corrected to zero each autumn before the first snowfall and acquire data in a 30 minutes time interval. Additionally, bi-weekly manual SWE observations are available from snow profiles. Here the water equivalent of the snow cover is obtained from multiple, seamless vertical coring using an aluminum cylinder with a cross-sectional area of $70 \, \text{cm}^2$ and a length of approximately 55 cm. The profiles are dug along transects north of the sub snow GPS station and thus their relative position changes over time. Furthermore, the sum of the measured





water equivalents of snowfall is used for comparison purposes. As the latter observations match the manual observations well over the three processed winter seasons, they are not shown in all figures of the present paper for clarity reasons.

The GPS derived SWE is compared to the snow pillow, snow scale, manual observations, as well as a combined reference (daily average of manual, snow pillow, and snow scale measurements). The combined reference is used as the three reference
sensors are not consistent to each other over the three seasons, which is discussed in Sect. 6.

## 3   SWE estimation model

A model is developed according to Steiner et al. (2018) to estimate the SWE above the sub snow GPS antenna. The model is based on the path delay of the GPS signals while propagating through the snow cover. Refraction at the air/snow interface and deceleration of the signal propagation velocity strongly depend on the amount and wetness of snow the signal has to travel
through. Similar to atmospheric refraction, the GPS signals received by an antenna under a snowpack are delayed by these refraction effects and lead to a longer electrical path through the snowpack (excess path length). The excess path length $\delta L_\mathrm{s}$ depends on the bulk refractive index $n_\mathrm{s}$ of the present snow, the incident angle $\nu_\mathrm{a}$, and the snow depth $d$:

$$\delta L_\mathrm{s} = d \cdot (\sqrt{n_\mathrm{s}^2 - \sin^2 \nu_\mathrm{a}} - \cos \nu_\mathrm{a}) \tag{1}$$

The derived model is, of course, a simplification as the snowpack consists not of one, but of several layers with differing path
delays. The model would allow to estimate the snow depth $d$ if the bulk refractive index $n_\mathrm{s}$ of the snowpack is known. Using the attenuation of the GPS signal strength would provide the snow wetness (Koch et al., 2014), which could be converted to the refractive index $n_\mathrm{s}$. However, the bulk density of snow $\rho_\mathrm{s}$ and the density of ice $\rho_\mathrm{i}$ is still needed in order to derive the SWE:

$$\mathrm{SWE} = d \cdot \frac{\rho_\mathrm{s}}{\rho_\mathrm{i}} \tag{2}$$

Because the bulk refractive index $n_\mathrm{s}$ and the snow density $\rho_\mathrm{s}$ are not known, the present study aims to use only the GPS path
delays to estimate the SWE directly. For that reason, an assumption is made: The GPS path delay in the layered snowpack is assumed to be nearly independent on snow density and the distribution of liquid water. Water has the highest impact on GPS observations within a snowpack (Steiner et al., 2018). Neglecting snow density and the water distribution, the model simplifies to a single water layer (Fig. 3) with the refractive index of water $n_\mathrm{w}$ (similar to tropospheric zenith delays, Hofmann-Wellenhof et al., 2001). The depth $d$ of the water layer corresponds thereby directly to the SWE (in mm w.e.):

$$\delta L_\mathrm{s} = \mathrm{SWE} \cdot (\sqrt{n_\mathrm{w}^2 - \sin^2 \nu_\mathrm{a}} - \cos \nu_\mathrm{a}) = \mathrm{SWE} \cdot F(\nu_\mathrm{a}) \tag{3}$$

where $F(\nu_\mathrm{a})$ can be interpreted as a mapping function to estimate the SWE of a snowpack (Sect. 5.1).

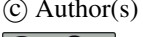



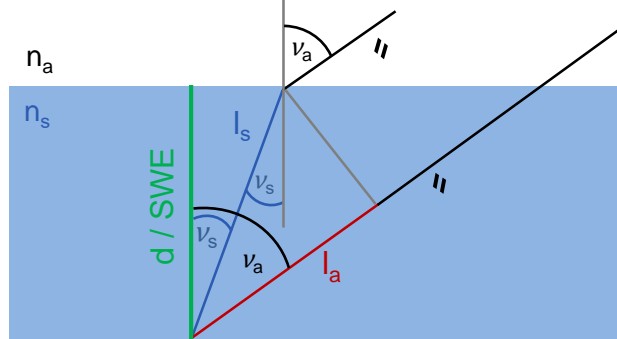

**Figure 3.** Geometry of signal paths in air ($l_\mathrm{a}$) if no snow is present and in a snowpack ($l_\mathrm{s}$) of depth $d$. $d$ corresponds to the SWE in case of the single water layer model assumption.

## 4 Method of investigation

The present study focuses on the use of GPS L1 data to allow a comparison to single frequency low cost systems in a future study. The GPS L1 data of the sub snow station is processed differentially to the base station data. The same equipment is used on both stations, eliminating the impact of antenna phase center offsets and variations. Atmospheric influences are mitigated

by the very short baseline of approximately 5 m.

The present study assumes a flat surface at the air to snow interface. This is an idealization as the roughness of the snow surface caused by environmental influences such as snow redistribution due to wind would change the signal path delays. However, the roughness effects at the Weissfluhjoch test site are considered to be of a small order and assumed to average out.

All data is processed over 24 hours (daily solutions) using the Bernese GNSS software (Dach et al., 2015) with sampling

data at 30 s. Observations from the snow free periods serve as GPS reference measurements without snow effects. Precise coordinates of the sub snow GPS station are computed as a combined solution of seven snow free reference days. The double difference processing is done for the very short baseline between the geodetic base and sub snow station. The GPS L1 processing utilized ionospheric, clock, and precise orbit products, provided by the International GNSS Service (Dow et al., 2009).

For the days, where snow fell on the antenna and the snow depth increased over the winter, the SWE is estimated as a daily solution using the derived model (Eq. 3). The corresponding time stamp is thereby set to noon as it can be interpreted as a daily average. The comparison to the automatic reference sensors (snow pillow and scale) is done on the daily average. Manual observations are usually carried out around 9 o'clock in the morning. The time shift in between the manual and GPS observations is neglected in the present study.



## 4.1 Snow water equivalent estimation

The error induced by the excess path length $\delta L_s$ (Eq. 3) when the signal propagates through a snow layer above the antenna, is introduced in the zero difference phase observation equation as an additional parameter $\delta L_s$:

$$L = \rho + \delta\rho + \delta L_s + \lambda N + \epsilon \qquad (4)$$

Thereby, $L$ is the observed path length, $\rho$ is the range between the sub snow antenna and a GPS satellite. All known path delays from the sub snow antenna to the GPS satellites (e. g., ionospheric and tropospheric delays) are included in $\delta\rho$. Hofmann-Wellenhof et al. (2001) thoroughly describes all known GPS path delays. The unknown number of ambiguities N is contained in $\lambda N$, with the GPS wavelength $\lambda$. The measurement noise is expressed as $\epsilon$.

The developed SWE estimation model was implemented directly into the Bernese GNSS software. As the SWE estimation is
expected to be highly correlated with other processing parameters such as ambiguities, station height, clocks, and troposphere parameters, the direct implementation permitted a processing, which takes these correlations into account.

By forming the single difference to the base station, the satellite clock, the atmospheric and relativistic effects cancel out at short baselines, in our case approximately 5 m. As the snow only affects the sub snow GPS station, the error introduced by the snow remains together with the receiver clock errors, ambiguities and multipath. The receiver clock errors can be eliminated by
double difference processing. The SWE parameter can be estimated, if the coordinates of the base and sub snow GPS station are precisely known and fixed. The daily solution processing provides one SWE estimate per day.

## 4.2 Sensitivity on GPS ambiguity resolution

The GPS phase observation equation includes the unknown ambiguity term $N$ (Eq. 4). The ambiguity is an integer number by definition as it refers to the counted number of full wave cycles between the receiver and a satellite. However, the estimated
ambiguity term is a float number due to instrument biases of the satellite and the receiver. The ambiguity term might be resolved to its integer value during the GPS processing. This ambiguity resolution leads to an accuracy increase, especially for small observation periods below one hour. The reduction of parameters in the GPS processing enhances the SWE parameter estimation. Successful ambiguity resolution depends on the satellite geometry (high number of observations, long observations periods), the baseline length (ionospheric and tropospheric refraction as well as orbit biases), and the multipath effects
(Hofmann-Wellenhof et al., 2001).

Not modelled error sources, such as the snowpack above the GPS antenna, degrades the ambiguity parameters. The subsequent systematic bias in the float ambiguity parameters leads to false resolved ambiguities and, if fixed for further processing, false estimated SWE. Due to the systematic bias in the float ambiguities, the ambiguities cannot just be rounded to the nearest integer. Sophisticated ambiguity resolution methods are used for the processing. Different GPS ambiguity resolution strategies
are therefore investigated in order to assess the effect of ambiguity resolution on SWE estimation:

   – **L1 float**: No ambiguity resolution. Estimate SWE with float ambiguities.



- **L1 fixed**: Resolve L1 ambiguities in a first step. Introduce the resolved ambiguities and estimate SWE in a second step

- **L1 SWE fixed**: Resolve L1 ambiguities and estimate SWE in one step

- **L5 fixed**: Resolve widelane (L5) ambiguities in a first step. Introduce the resolved ambiguities and estimate SWE in a second step. These results are not plotted for visibility reasons, however match the behaviour of the L1 fixed solutions.

- **L5 SWE fixed**: Resolve L5 ambiguities and estimate SWE in one step

The **widelane linear combination (L5)** is expected to enhance the ambiguity resolution. It is a linear combination of the L1 and L2 observations and is defined as:

$$L_5 = \frac{f_1}{f_1 - f_2} L_1 - \frac{f_2}{f_1 - f_2} L_2 \tag{5}$$

with the two frequencies $f_1 = 1575.42\,\mathrm{MHz}$ and $f_2 = 1227.60\,\mathrm{MHz}$. The widelane wavelength $\lambda_\mathrm{w}$ of 86 cm is significantly

larger than the L1 wavelength ($\lambda = 19\,\mathrm{cm}$), allowing a better separation of the SWE and the ambiguity parameters.

$$\lambda_\mathrm{w} = \frac{c}{f_1 - f_2} = 86\,\mathrm{cm} \tag{6}$$

## 5    Results

Sect. 5.1 describes the GPS derived SWE estimation results. The dependency on the GPS processing, especially ambiguity resolution techniques, are illustrated further on.

From end of April to beginning of May 2016, no data are available due to a storage failure caused by internet connectivity break down. A loss of lock in mid of April 2018 resulted as well in missing data of the sub snow GPS station. The receiver was restarted at the end of June 2018, however, after the snow had melted completely.

### 5.1    Snow water equivalent estimation

The SWE is estimated based on an ambiguity float solution (L1 float) using the derived model (Sect. 3). The time series of

the sub snow GPS derived SWE estimations for the 2015/16 – 2017/18 seasons is shown in Fig. 4 (top). Generally, the SWE derived from sub snow GPS corresponds well to the reference sensors over the three seasons. However, the GPS derived SWE corresponds inconsistently to each of the reference sensors as illustrated by the difference ($\Delta$SWE) of the GPS derived SWE to the individual reference sensors (Fig. 4, middle). The GPS derived SWE fits very accurately to the snow scale during the dry snow period in 2015/16. Later on, at the beginning of the melting season, the GPS derived SWE fits in between the snow

pillow and snow scale observations and fits very accurately to the manual observations.

The sub snow GPS system seems to overestimate the SWE from January to May 2017, compared to all three reference sensors, but fits best to the snow pillow observations until April 2017. In the following melting period, the GPS derived SWE





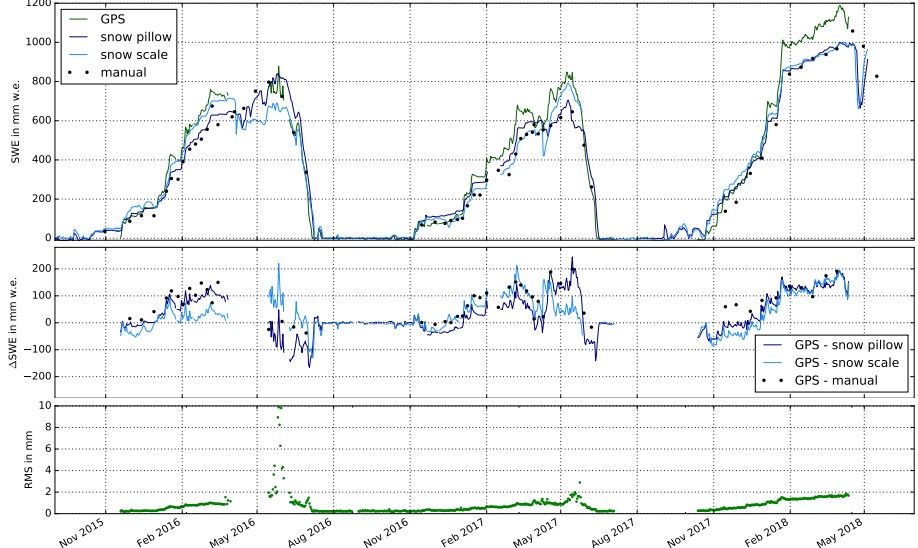

**Figure 4.** Time series of the GPS derived SWE estimations and the reference sensors (top), the differences to the three reference sensors (middle), and the SWE estimation RMS (bottom) of the GPS float solution for the 2015/16 − 2017/18 seasons.

corresponds accurately to the snow scale observations. The same behaviour is observed until January 2018. Afterwards, the SWE is again overestimated by the sub snow GPS system. This could be due to an uneven snow distribution, which is not captured by the model yet. Additionally, deviations between the snow pillow and snow scale in wet snow conditions are present at the Weissfluhjoch test site for each melting period. Fig. 4 (bottom) shows the RMS of the sub snow GPS derived

daily SWE estimations from the Bernese GNSS software output, over the three seasons. The SWE is estimated with a RMS of approximately 1 mm w.e. in average. An increase of the RMS with SWE is seen during each season. The high RMS values in May 2016 are caused by a significant reduced number of observations in this time period (Sect. 7).

Fig. 5 shows the regression analysis for the sub snow GPS derived SWE estimations with the reference sensor's SWE observations for the 2015/16 − 2017/18 seasons. The regression analysis is calculated with respect to a) the combined reference,

b) manual, c) snow pillow, and d) snow scale observations. The regression coefficients and additional statistics are listed in Table 1. Time periods without snow above the sub snow GPS antenna are thereby excluded.

The sub snow GPS derived SWE is highly correlated with a cross correlation coefficient ($cc$) of 0.99 to all reference sensors and 0.98 to the manual observations. Please note that the manual observations are more sparse and thus have much less values to compare with (number of samples $n$, Table 1). The regression slopes ($m$) underline close agreement between the GPS derived

SWE and the reference sensor measurements. The sub snow GPS overestimates the SWE compared to the manual observations with an offset ($b$) of 21 mm. The GPS derived SWE are biased by -28, -27, and -31 mm w.e. w.r.t. the snow pillow, snow scale, and combined reference measurements, respectively.

Generally, the GPS derived SWE shows a good agreement over the three full seasons to the reference sensors with an RMSE of 70 mm w.e. to the combined reference and the snow scale observations. The RMSE increases to 77 mm w.e. for the




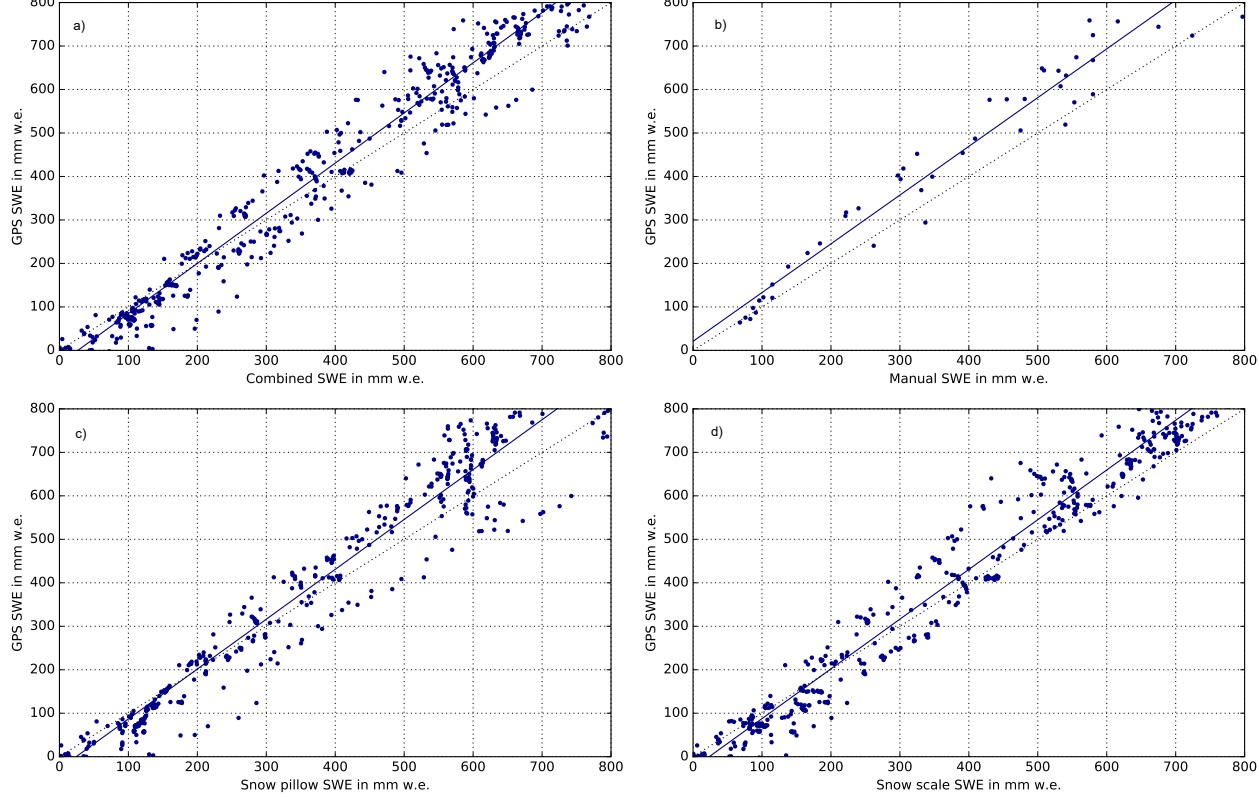

**Figure 5.** Regression analysis of SWE estimation from GPS float solution to a) the mean of the three reference sensors, b) the manual observations, c) the snow pillow, and d) the snow scale for the 2015/16 – 2017/18 seasons. The black dotted line represents the 1:1 line.

snow pillow and 95 mm w.e. for the manual observations. Best agreement is demonstrated to the snow scale with a median relative bias (MRB) of 3.5 % over 664 samples, followed by the snow pillow with a MRB of 10.1 % over 680 samples. The manual observations fit least to the GPS derived SWE with a MRB of 19.1 %, however only 51 samples are available for this comparison. Nevertheless, the GPS derived SWE agrees well to the combined reference with a MRB of 8.5 % over 685 samples

5    for the seasons 2015/16 – 2017/18. A MRB below 10 % is considered a good agreement as each reference sensor is prone to errors, which is discussed in Sect. 6.

The statistics of the comparison to the combined reference are also listed in Table 1 for each individual season 2015/16, 2016/17, and 2017/18 with 204, 299, and 182 samples per season, respectively. Best results are obtained for the 2015/16 season with a correlation coefficient of 0.99, a MRB of 1.4 %, a regression slope and offset of 1.1 and -24 mm w.e., and a RMSE of

10    52 mm w.e. Season 2016/17 shows the least agreement in terms of the MRB of 12 %. The regression statistics illustrate good agreement as well with a $cc$ of 0.99, a regression slope and offset of 1.2 and -16 mm w.e., and a RMSE of 61 mm w.e. Results of the 2017/18 season agree well with a MRB of 9.6 %, a $cc$ of 1.0, a regression slope and offset of 1.2 and -84 mm w.e., and a RMSE of 97 mm w.e.





**Table 1.** Regression coefficients and additional statistical values for the comparison of the sub snow GPS (L1 float solution) with each reference sensor. "Combined" indicates the average of the manual, snow pillow, and snow scale observations. $b$ and $m$ are the offset and slope of the regression line, $cc$ is the correlation coefficient, MRB the median relative bias, and $n$ the number of samples.

| Season | Sensor | $b$ in mm w.e. | $m$ | $cc$ | RMSE in mm w.e. | MRB in % | $n$ |
|---|---|---|---|---|---|---|---|
| | Manual | 21 | 1.1 | 0.98 | 95 | 19.1 | 51 |
| 2015/16 – 2017/18 | Snow pillow | -28 | 1.1 | 0.99 | 77 | 10.1 | 680 |
| | Snow scale | -27 | 1.1 | 0.99 | 70 | 3.5 | 664 |
| | **Combined** | **-31** | **1.2** | **0.99** | **70** | **8.5** | **685** |
| 2015/16 | Combined | -24 | 1.1 | 0.99 | 52 | 1.4 | 204 |
| 2016/17 | Combined | -16 | 1.2 | 0.99 | 61 | 12.0 | 299 |
| 2017/18 | Combined | -84 | 1.2 | 1.0 | 97 | 9.6 | 182 |

## 5.2 Sensitivity on GPS ambiguity processing

The SWE estimations derived from the sub snow GPS L1 float solution agree well with the reference sensors (Sect. 5.1). The SWE estimations are, however, assumed to depend strongly on the GPS processing, especially the GPS ambiguity resolution. The SWE estimation sensitivity on the ambiguity resolution techniques is therefore further investigated for the strategies

described in Sect. 4.2: L1 float, L1 fixed, L1 SWE fixed, and L5 SWE fixed.

Fig. 6 illustrates the effect of the different ambiguity resolution techniques on the SWE time series derived from the sub snow GPS system. The time series are compared to the combined reference observations (daily average of snow pillow, snow scale, and manual observations). The L1 float, the L1 SWE fixed, and the L5 SWE fixed solutions agree with each other over the 2015/16 – 2017/18 seasons. All three strategies overestimate the SWE over the three seasons compared to the combined

reference. The L1 float and L1 SWE fixed solutions fit thereby best to the combined reference.

On the other side, the L1 fixed solution deviates significantly from the combined reference in all three seasons, especially during the beginning of the melting seasons and the ablation periods (Fig. 6, top and bottom). The L5 fixed solution shows the same behaviour, although not plotted in the present paper for visibility reasons. Both methods resolve the GPS ambiguities in a first step and estimate the SWE in a second step with introducing the fixed ambiguities as known. These methods underestimate

the SWE in all three seasons in the order of half to a full wavelength (about 19 cm for GPS L1). The bias is thus assumed to be caused by a false ambiguity resolution. A constant part of the SWE is absorbed by the ambiguity parameters. This can occur if the excess path length caused by the overlying SWE is the same as the GPS wavelength. This behaviour is shown for all three seasons for the L1 fixed solution at a SWE higher than approx. 200 mm w.e. The SWE estimation is thus biased as the false resolved ambiguities are fixed in all following processing steps. Note, that the L5 fixed solution has a very low MRB. This is

due to an overestimation of SWE in the accumulation period, and a stronger underestimation in the ablation period resulting in low values in average.

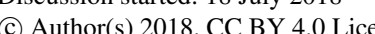



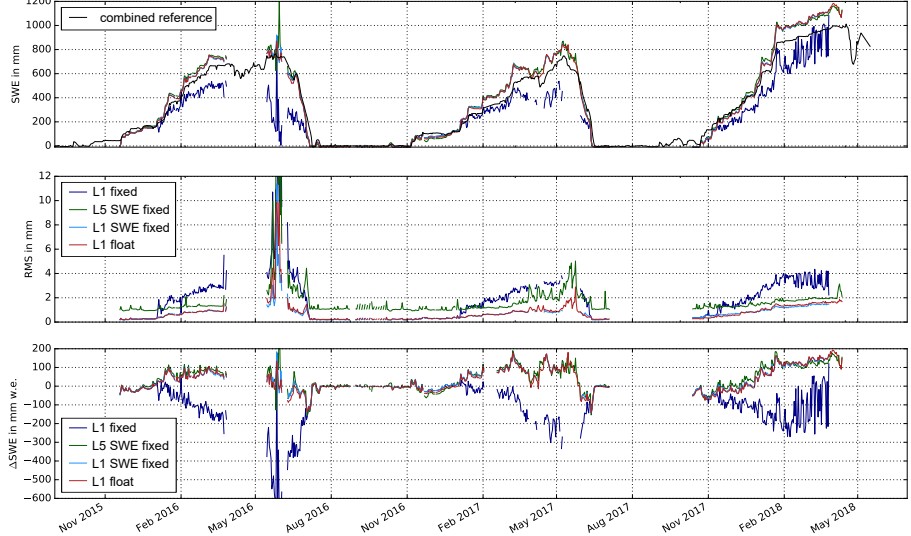

**Figure 6.** GPS based SWE estimation (top) and its RMS (middle) from different ambiguity resolution strategies. The black line shows the mean SWE of the three reference sensors. The bottom plot shows the differences of GPS based SWE estimation from different ambiguity resolution strategies to the mean SWE of the three reference sensors.

The combined estimation of the SWE and the ambiguity parameters in one step improves the results significantly (L1 SWE fixed and L5 SWE fixed). The correlations of the SWE and the ambiguity parameters are calculated exemplary for one day with high SWE (approx. 700 mm w.e. on 24 May 2017). The SWE and ambiguity parameters are thereby not correlated with correlations of 0.2 in average, allowing a good separation of the SWE and ambiguity parameters in the combined processing.

Fig. 6 (middle) shows the RMS of the different ambiguity resolution techniques. The L1 float and the L1 SWE fixed solutions estimate SWE most accurately with an RMS around 1 mm w.e. The float solution is thereby more noisy. The L5 SWE fixed solution has a higher RMS caused by the approx. 5.7 times higher noise of the widelane linear combination (Hofmann-Wellenhof et al., 2001). All three strategies show a seasonal trend, correlated to the SWE. The RMS of the L1 fixed solution is highest and most noisy. Problems occured in May 2016, where all solutions except the float solution show high noise in the

time series (SWE and $\Delta$SWE) and the RMS. This is caused by a significant reduced number of observations in this time period (Sect. 5.3).

The regression coefficients and additional statistics for the different ambiguity resolution techniques compared to the combined reference are listed in Table 2 for the 2015/16 – 2017/18 seasons. Detailed results from the L1 float solution are summarized in Table 1. The sub snow GPS derived SWE from the L1 float, L1 SWE fixed, and the L5 SWE fixed solutions is highly

correlated to the combined reference with a $cc$ of 0.99 and a MRB of 8.5 %, 8.0 %, and 11.4 %, respectively. The L5 fixed solution has a $cc$ of 0.98 and the lowest MRB of 5.4 %. The RMSEs of the L1 float (70 mm w.e.), L1 SWE fixed (66 mm w.e.), the L5 fixed (72 mm w.e.), and the L5 SWE fixed (75 mm w.e.) solutions are approximately equivalent. The slope of the regression analysis is 1.1 for the L1 SWE fixed and L5 fixed solution, and 1.2 for the L1 float and L5 SWE fixed solutions. The GPS





**Table 2.** Regression coefficients and additional statistical values for the comparison of different sub snow GPS processing strategies with the combined reference sensors (average of the manual, snow pillow, and snow scale observations) for the $2015/16 - 2017/18$ seasons. $b$ and $m$ are the offset and slope of the regression line, $cc$ is the correlation coefficient, MRB the median relative bias, and $n$ the number of samples. The lines in bold indicate the solutions which fit thereby best to the combined reference (L1 float and L1 SWE fixed) using only single frequency signals.

| GPS processing | $b$ in mm w.e. | $m$ | $cc$ | RMSE in mm w.e. | MRB in % | $n$ |
|---|---|---|---|---|---|---|
| **L1 float** | **-31** | **1.2** | **0.99** | **70** | **8.5** | **685** |
| L1 fixed | -9 | 0.8 | 0.92 | 154 | 23.0 | 587 |
| **L1 SWE fixed** | **-26** | **1.1** | **0.99** | **66** | **8.0** | **633** |
| L5 fixed | -15 | 1.1 | 0.98 | 72 | 5.4 | 685 |
| L5 SWE fixed | -23 | 1.2 | 0.99 | 75 | 11.4 | 685 |

derived SWE is overestimated with all ambiguity resolution techniques with an offset of -15 mm w.e. (L5 fixed), -23 mm w.e. (L5 SWE fixed), -26 mm w.e. (L1 SWE fixed), and -31 mm w.e. (L1 float). The L1 SWE fixed solution is favourable in terms of statistics and agreement to the combined reference, followed by the L1 float solution. However, systematic and stochastic effects in the GPS processing residuals (Sect. 5.4) are still present for the L1 float and L1 SWE fixed solution in all three
seasons. The L1 fixed solution is least correlated with a $cc$ of 0.92 to the combined reference and has the highest MRB and RMSE of 23 % and 154 mm w.e., respectively. The regression analysis of the L1 fixed solution has a low offset of -9 mm w.e. and a slope of 0.8. Less samples (587) are a result of an unsuccessful ambiguity resolution.

**5.3    GPS processing properties**

Fig. 7 shows different GPS processing properties: sigma a posteriori ($\sigma_{\mathrm{post}}$, top) of the least squares GPS processing and
number of ambiguity parameters (middle) for different ambiguity resolution techniques during the differential GPS processing over the $2015/16 - 2017/18$ seasons. The number of observations for the differential GPS L1 processing are illustrated in Fig. 7 (bottom). Around 20'000 daily observations are possible at the Weissfluhjoch test site. A high amount of liquid water in the snowpack attenuates the GPS signals strongly (Steiner et al., 2018). The reduced signal strength results in a frequent loss of lock to the satellites and less observations. The reduced number of observations are visible in all three years during the melting
periods and depend on the liquid water content and the receiver tracking threshold. After the data storage failure in April 2016, the number of observations drops significantly due to the wet snowpack and increases very slowly with decreasing SWE. The antenna connector of the sub snow GPS antenna was nearly broken in September 2016, causing a high noise in the observations number in this time period. The connector was changed and the number of observations was again stable around 20'000 in the end of September 2016.


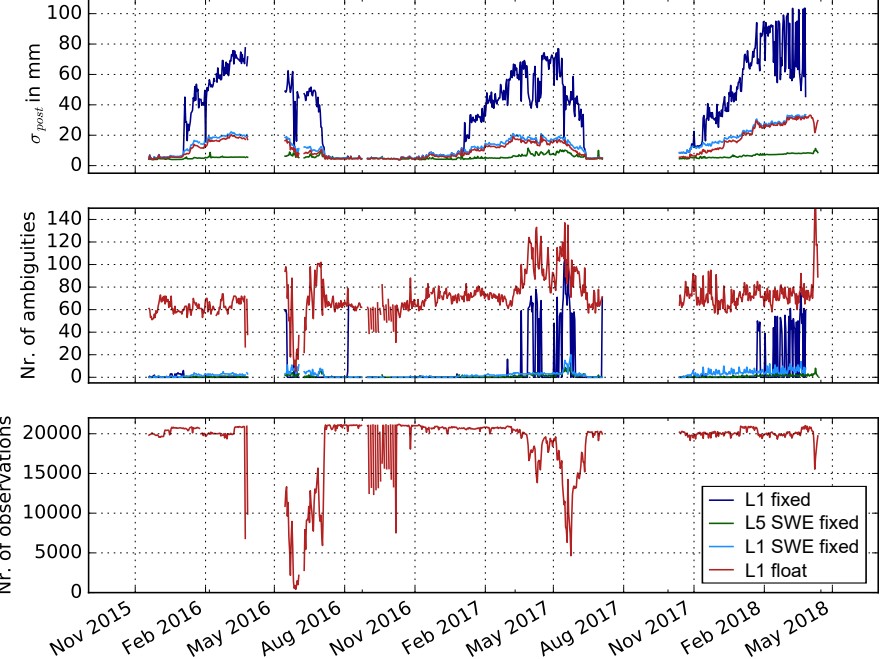

**Figure 7.** Properties of the sub snow GPS processing: sigma a posteriori ($\sigma_{\mathrm{post}}$, top), number of ambiguity parameters (middle), and number of observations (bottom) for different ambiguity resolution techniques.

Fig. 7 (top) shows the sigma a posteriori ($\sigma_{\mathrm{post}}$) of the GPS processing from the Bernese GNSS software for the different ambiguity resolution techniques described in Sect. 4.2. The L1 fixed solution is least accurate with a maximum $\sigma_{\mathrm{post}}$ of 100 mm. The L1 fixed solution has a high noise, especially during the beginning of the melting period 2018 and shows strong seasonal trends. The L1 float and the L1 SWE fixed solutions agree with each other, with a maximum $\sigma_{\mathrm{post}}$ of 35 mm in April

2018. A seasonal trend is visible for all three years. The L5 SWE solution shows lowest $\sigma_{\mathrm{post}}$ of maximally 10 mm and no significant seasonal trend. A larger number of parameters (more observations of the L5 solutions due to L1 and L2 observations or included ambiguity parameters in the L1 float solution) reduces the $\sigma_{\mathrm{post}}$ significantly due to increased redundancy.

The number of ambiguity parameters is illustrated in Fig. 7 (middle) before ambiguity resolution (L1 float solution) and after (L1 fixed, L1 SWE fixed, L5 SWE fixed). Approx. 70 ambiguities are set up in average for each day of the 2015/16 –

10 2017/18 seasons. The number of ambiguities increases for days with a strong melt. This is caused by a the strong attenuation of the GPS signals when a high amount of liquid water is present in the snowpack. The resulting loss of lock to the satellites forces new ambiguity parameters. The number of ambiguities decreases, of course, in case of reduced number of observations (Fig. 7 , bottom), e. g. in May – June 2016. Higher number of ambiguities are, however, seen in June 2017. A huge amount of fragmented observations results in more ambiguity parameters. All ambiguity parameters should be resolved and fixed correctly

after the successful ambiguity resolution step in the GPS processing. The L5 SWE solution resolves almost all ambiguities. This confirms the assumption of better ambiguity resolution using the widelane linear combination due to the better separation





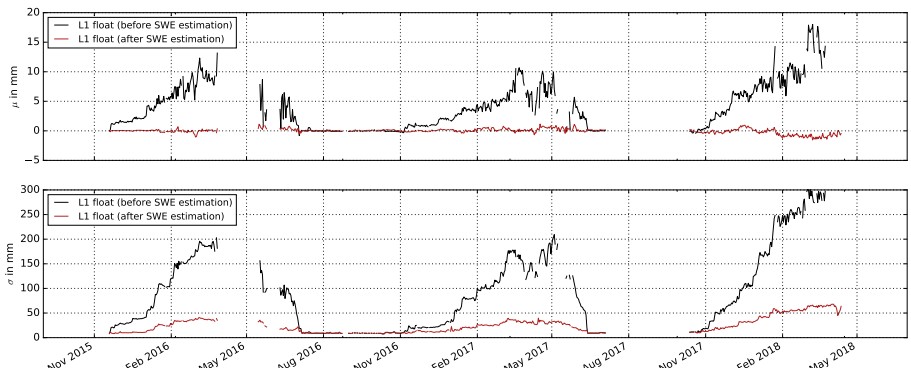

**Figure 8.** Mean ($\mu$) and standard deviation ($\sigma$) of the GPS double difference residuals from the L1 float solution before (black) and after (red) SWE estimation.

of ambiguity and SWE parameters caused by the large wavelength of 86 cm. In the L1 SWE fixed solution, not all ambiguities could be resolved, especially in the beginning of the melting periods. The most significant behaviour is, however, shown for the L1 fixed solution. Thereby, no significant amount of ambiguities (around 50 ambiguities) could be resolved during melting periods in 2016/17 or periods of a high SWE above 800 mm w.e. in the 2017/18 season.

## 5.4 GPS processing residuals

The GPS processing residuals provide information on the applicability of the developed model to the SWE estimation. The GPS processing residuals should be normally distributed around zero without error influences, such as the overlying snowpack. The snowpack above the GPS antenna deteriorates the GPS observations based on the path delay of the GPS signals (Sect. 3). This effect should induce a significant seasonal effect in the GPS double difference residuals (further called GPS processing residuals) if no model is applied in the processing. Fig. 8 shows the daily mean ($\mu$) and standard deviation ($\sigma$) of the GPS processing residuals from the L1 float solution over the 2015/16 − 2017/18 seasons. The mean of the residuals illustrates systematic effects, whereas the standard deviation points out stochastic effects. The black line indicates no applied model and the red line illustrates the residuals if the SWE estimation model is applied. Significant systematic and stochastic effects show up in the mean and standard deviation if no model is applied. Both effects follow the seasonal SWE development with a maximum around 18 mm in the mean, and 300 mm in the standard deviation.

The systematic trend in the mean of the GPS processing residuals is eliminated by applying our derived model. Noise in the order of 1 mm in average is still present, especially in 2018, where the SWE was above average. The stochastic effects in the standard deviation are significantly reduced to 40 mm in average. The model is thus able to correctly estimate the SWE as the effects in the residuals due to the overlying snowpack are significantly reduced. The remaining noise could be due to the assumption of a flat surface at the air/snow interface, the neglected roughness or layered snowpack (Sect. 4 and 3).



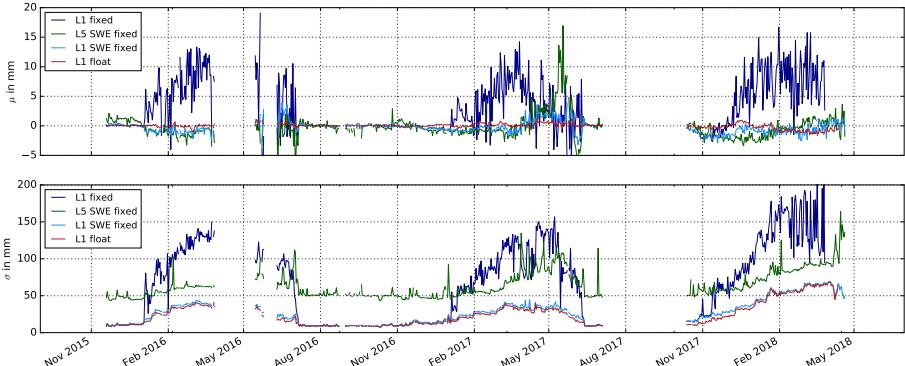

**Figure 9.** Mean ($\mu$) and standard deviation ($\sigma$) of the GPS double difference residuals after SWE estimation for different ambiguity resolution techniques.

Fig. 9 illustrates the influence of the different ambiguity resolution techniques on the daily mean ($\mu$) and standard deviation ($\sigma$) of the GPS processing residuals. The L1 float solution is already described for Fig. 8. The L5 SWE solution agrees with the L1 SWE fixed solution in the daily mean (Fig. 9, top) with small deviations. Large deviations are present in May 2017. Both techniques model the systematic effect of the overlying snowpack quite well, with a remaining noise around 3 mm. The standard

deviation of the L1 SWE solution performs almost similar to the L1 float solution for all seasons. The standard deviation (Fig. 9, bottom) of the L5 SWE solution is approx. 5.7 times higher than the L1 float solution due to the increased noise level of the linear combination. The residuals for the L1 fixed solutions are significantly higher with maximal values reaching 20 mm in the mean and 200 mm in the standard deviation of the GPS processing residuals. About 70 % of the snowpack effect remains over the three seasons compared to the solution without an applied model (Fig. 8). This suggests a wrong ambiguity fixing in the

first step, leading to a strong weakness of the derived model to estimate the SWE correctly when using the L1 fixed processing strategy.

# 6   Discussion

## 6.1   Comparison with other measurements

Generally, the sub snow GPS derived SWE estimations correspond well to the reference sensors with a cross correlation

coefficient ranging from 0.97 (manual) to 0.99 (snow pillow, snow scale, combined reference) over all three seasons. All reference sensors are prone to errors and SWE observations are not consistent with each other, resulting in a lack of real ground truth data to compare with. The snow pillow and snow scale have a larger measurement area than a manual point sample, whereas the sub snow GPS uses signals arriving at different line of sights within an area of approximately 5 m. The three reference sensors are located at different locations within the test site (Sect. 2) and the manual observations cannot




be taken exactly at the same spot each time due to its destructive method. The manual SWE observations were, however, converted to one observer location in the middle of the test site. The snow pillow and snow scale measure at different locations, approximately 20 m West next to the sub snow GPS station, resulting in small uncertainties.

The snow depth is observed to vary about 10 cm spatially within the test site. This variation is less than 10 % of the seasonal
snow depth except at the beginning and end of the three snow seasons. The maximal snow depth was approximately 1.7 m in 2015/16 and 2016/17,and approximately 3 m in 2017/18. A larger snow depth at the sub snow GPS site could explain a small overestimation of GPS derived SWE. Ideally, co-located SWE data should thus be used to evaluate the derived SWE estimation results. Although, this would reduce the influence of the unequal snow distribution, the sensor biases are, however, still remaining. Manual SWE observations have an uncertainty of about 10 % due to the within site variability of snow density
(Jonas et al., 2009). Existing ice layers or percolated liquid water complicates the snow sampling (Smith et al., 2017).

Snow weighting sensors like snow scale and snow pillow experience measurement artefacts (e.g. bridging effects), especially at the beginning of the melt season. Snow pillows are usually less affected than snow scales due to the larger surface (Beaumont, 1966). Furthermore, melt water percolates through the snowpack towards the ground, forming a basal liquid water or ice layer, infiltrating into the soil, or percolating out of the measurement area. These effects can cause SWE over- or underestimation by
the snow pillow, snow scale, or manual observations. A basal melt water layer still delays the GPS signals and could contribute to the overestimation of SWE in the beginning of the melting periods, increasing the bias between the GPS and reference SWE observations.

## 6.2   Sensitivity on GPS ambiguity processing

The SWE estimation based on the sub snow GPS system is highly sensitive to the GPS ambiguity resolution techniques in
the phase based double difference GPS processing (Sect. 5.2). A succesfull SWE estimation is possible by simultaneously resolving the GPS ambiguities (L1 SWE fixed solution). Ambiguity resolution in a separate processing step (L1 fixed solution) before SWE estimation with the introduced fixed ambiguities leads to less resolved ambiguities or false ambiguity resolution and thus biased SWE. This is especially the case for SWE higher than one wavelength ($\lambda = 19$ cm), as the path delay induced by the SWE is partly compensated by the ambiguity parameters. The error in the SWE can thereby reach 200 mm w.e. Using
a larger wavelength of the L5 SWE solution facilitates the separate SWE and ambiguity estimation, compared to the L1 fixed solution. However, the measurement noise is significantly increased when using a linear combination and multiple frequencies are required.

The L1 float and L1 SWE fixed solutions performed best overall parameters. The SWE is estimated with the lowest RMS, agrees best with the reference sensors (lowest RMSE, MRB, and highest $cc$), and least systematic and stochastic effects are
present in the GPS residuals. The L1 SWE fixed solution performs slightly better than the L1 float solution in terms of the statistical values. The L1 float solution is, however, chosen as favourable SWE estimation strategy as no ambiguity resolution is required. Thereby, the processing is simplified significantly and false ambiguity resolution is prevented. Moreover, only single frequency data is needed for the L1 float or L1 SWE fixed processing strategies, allowing cheaper equipment and a faster processing.





### 6.3 Advantages and limitations

The sub snow GPS system provides a new and promising method for daily SWE quantification. The method is easy to install, requires few maintenance, is non destructive, and provides automatic observations with a high temporal resolution. Due to the small size, the system could be even installed in mountain slopes. No access is required during the snow period and this method could therefore be applied, for example, in avalanche prone terrain. Power supply of the GPS receiver can be a limiting factor in these areas. Loss less cables from the sub snow GPS antenna to the receiver, however, allow the GPS receiver and power supply installation outside the avalanche terrain.

Small SWE (10 mm w.e.) could already be quantified from early snow falls, e. g. October 2016. Precise SWE estimations were possible until 800 mm w.e. in 2016 and 2017. SWE above 800 mm w.e. (2018) are overestimated by the sub snow GPS SWE estimation model, compared to the reference sensors. The overestimation could be explained by an uneven snow distribution within the test site, which is not yet captured by the model. The upper limit is not assessed yet. Melted snow percolated to the ground, forming a liquid water layer above the sub snow GPS antenna could prevent GPS signal reception if deeper than 35 mm (GPS signal penetration depth in liquid water around 0 °C, Steiner et al., 2018).

### 6.4 Representativeness

The present study is carried out over three snow seasons in the Alps at a high altitude. The study should thus be representative for an Alpine snowpack at similar altitudes, as a large amount of data is analyzed, including melting seasons. Using longer baselines between the GPS reference station and the sub snow GPS antenna with a large difference in elevation could complicate the SWE estimations due to tropospheric refraction. The derived sub snow GPS model should be as well representative for a polar snowpack as it is usually dry. Problems could be caused by the GPS satellite distribution with missing satellites in the North. Adding more satellite systems could increase the observations and improve SWE estimation analogously. The presented method is, however, not representative for forested areas, as satellite visibility is very limited and the GPS phase signals are highly attenuated, delayed, or obstructed by the trees. A good GPS visibility is important at all locations and should be considered, especially in narrow mountain areas.

### 7 Conclusions

The newly developed model is applied to a seasonal snowpack in order to investigate the potential of using GPS L1 observations from a geodetic GNSS system for daily SWE estimation. Sub snow GPS is a promising method for point wise SWE quantification. The snowpack is not destroyed or disturbed due to the automated, continuous, self sustainable observation method and the effort for installation is relatively small. Remote (on line) access is possible and almost no maintenance is required for the small sized equipment. The presented model enables the direct estimation of SWE if both the reference and submerged station coordinates are precisely known. The use of a single water layer model for SWE quantification is encouraged as the SWE depends on the bulk relative permittivity in the snowpack. SWE could be estimated with a relative bias below 10 % compared



to a combination of three independent reference sensors (snow pillow, snow scale, and manual observations). The proposed method successfully estimated SWE over three full seasons, including ablation periods.

The assumption of GPS ambiguity resolution as a critical parameter for GPS observations within a snowpack can be confirmed. False ambiguity resolution biases the SWE estimation up to 200 mm. It is shown to be important to resolve the ambiguities in one step together with the SWE estimation, instead of separately. In any case, an ambiguity float solution performs better than using false resolved ambiguities.

The promising results of this study encourage the assessment of a low cost GPS system and the comparison with the geodetic equipment in a next step. The potential of sub daily SWE estimation will be evaluated further on.

*Data availability.* All data sources are given in Sect. 2. The analyzed data were gathered in the frame work of a Ph.D. project funded by the SNSF. The data are stored at ETH and are available by request.

*Author contributions.* L. Steiner designed the study, performed data analyses and prepared the manuscript. M. Meindl implemented the SWE model into the GNSS processing software. C. Fierz provided in-depth knowledge of snow science, reference data, and access to the test site. A. Geiger supervised the study.

*Competing interests.* The authors declare no competing interests.

*Acknowledgements.* This project was supported by the Swiss National Science Foundation (SNF 200021 156867). We like to thank Bernhard Richter, GNSS Business Director at Leica Geosystems AG for the support of this project by providing two geodetic GNSS antennas (AS 10) and receivers (GR 10). We thank Christoph Marty for reviewing the manuscript, providing ground truth data, and helpful discussions. Thanks to Marc Rüesch, technical employee of the WSL Institute for Snow and Avalanche Research SLF, for a great support in all questions related to the test site.



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
