# Peer review of "An assessment of sub snow GPS for quantification of snow water equivalent"

_The Cryosphere, 2018_

## Referee Comment (RC1) · Anonymous Referee #1 · 13 Aug 2018

This study studies snow water equivalent (SWE) from comparing GPS signals between a free receiver and one nearby buried under snow. This study is very well designed, conducted and presented. Very useful results! I have only a few minor suggestions:

(1) Figs 1 and 2 show a lot of metal around the measurement site. A few words about (potential) disturbance of the (differential) GPS signals might be useful.

(2) page 6, line 19: The time shift between the manual and GPS observations is neglected, but might be substantial for warm/melting conditions. Add some evaluation of this effect.

(3) page 8, line 2: explain the bold titles and text a bit better. For instance I don't understand why the 2nd method is called SWE fixed, if SWE is estimated.

[Figure]

(4) page 9, line 18 and following. Isn't there an effect to be expected on the SWE estimate whether the snow is dry, or wet (melting). Do you still expect a continuous SWE estimate even for this phase transition? Something in the data about that and discussion of it?

---

## Referee Comment (RC2) · Anonymous Referee #2 · 23 Aug 2018

The presented paper shows an interesting and well designed study to highlight the strength and shortcomings of using GPS measurements from sub snow antennas to estimate the snow water equivalent. Nearly three years of GPS data together with reference data are used for this task. This paper brings into application the method described in Steiner et al. (2018, J.Geodesy) where the model of a thin water shell was developed to describe the snow water equivalent. The scope of the study fits the topics of this journal. The paper contains significant new material and interesting results. It is very well structured and thus easy to follow.

I have some smaller remarks: The reviewer would like to point the attention of the author to the former IAG study group on site specific effects where among other the impact on snow on the radoms and antennas were studied. Furthermore the following

two references could be useful: Jan M. Johansson: Special Study Group 1.158: GPS Antenna and Site Effects. Two further papers could be useful references: S. Vey, A. Güntner, J. Wickert, T. Blume, H. Thoss and M. Ramatschi, "Monitoring Snow Depth by GNSS Reflectometry in Built-up Areas: A Case Study for Wettzell, Germany," in IEEE Journal of Selected Topics in Applied Earth Observations and Remote Sensing, vol. 9, no. 10, pp. 4809-4816, Oct. 2016.doi: 10.1109/JSTARS.2016.2516041 S. Tabibi, F. Geremia-Nievinski and T. van Dam, "Statistical Comparison and Combination of GPS, GLONASS, and Multi-GNSS Multipath Reflectometry Applied to Snow Depth Retrieval," in IEEE Transactions on Geoscience and Remote Sensing, vol. 55, no. 7, pp. 3773-3785, July 2017.doi: 10.1109/TGRS.2017.2679899

Eq.(1) For curiosity: it would be nice to give the typical range of delays.

P7.l11 "which takes these correlations into account". Please specify more clearly what is the meaning of sentence and how this is realized when a common estimation is carried out.

P7.l16 Please could you give a comment on the potential maximum temporal resolution of SWE estimates. Which SWE data rate could reliably be feasible and are there useful applications, such as monitoring intense snow fall during a day or extreme melting?

Table 1. For me, some more explanation how to read the numbers are necessary. Are the values of the first lines (2015/16-2017/18) to combine? If so, please explain how to interpret the number of samples and why b is smaller than all the other values while m is only slightly steeper. Is a weighted average used when combining the "combined" individual years to the overall "combined" solution?

Figures 4, 6, 7, 8, 9: If not regulated differently by the journal style file. I would personally prefer increased figure sizes with a larger line size and a larger caption font size for a better readability.

P14.l1ff I wonder, the large a posteriori variance factor, especially when comparing

to the expected noise of L1 observations of 1-2 mm. My explanation is rather that systematic effects remain in the residuals that yield an increase in the a posteriori variance factor, see also your explanations on P16.l9ff. Do you have any explanation for this problem? We can suppose the antenna positions and the relative distance well known.

P18. Could you give a typical snow volume, the SWE is representative for?

P 21 l22 Please check the reference Rao et al.

---

## Short Comment (SC1) · 25 Aug 2018

Congratulations on this well-structured nice work – we followed your results with great interest. It is positive that research on submerged antennas for the derivation of snow properties is increasing – this article is a further step on this topic. The idea of using a widelane linear combination for data from geodetic receivers is interesting.

Thank you for citing our research (Henkel et al. 2018 and Koch et al. 2014). It would be great, if you could insert in your introduction that our study on GNSS SWE derivation was also conducted at the study site Weissfluhjoch. Would be interesting to compare our results in future. Maybe you can also refer to our study at the point you are describing the method that you also applied one variant estimating L1-data ambiguities

and snow parameters in one step as we did this as well in Henkel et al.

Regarding page 5, line 23, we have one suggestion as the seperation between dry and wet snow is an issue: You write that SWE can be estimated by a single water layer using the refractive index of water. This might be approx. true for wet snow. However, if the snow is dry, it would be rather a single layer of ice (e.g. compressing all snow particles to one thin ice layer) which would have a quite different refraction index (as you also demonstrated in your former paper). The different refraction would then also have an impact on the excess path length (ice vs water). Maybe you can assume an average refraction index for both cases (dry and wet snow) or just mention that this point might be a source of error and discuss the phase change from dry to wet snow a bit more in detail.

Best regards

---

## Author Comment (AC1) · 12 Sep 2018

Thank you for your interest in our research work and your comments. A comparison of our methods and results would indeed be very interesting.

The Widelane linear combination is assessed together with different processing methods in order to investigate the impact of the GPS processing on the SWE estimation. However, we recommend to use the L1 SWE fixed or L1 float method as only single frequency data is needed. Thus, our method would also be well suited for low-cost GPS systems.

We added a sentence about your study site in the introduction.

Regarding the L1 SWE fixed processing: As we understand from Henkel et al. (2018),

you process the GPS data in standard double difference mode, estimate and fix the ambiguities, and compute observation residuals. The SWE is then estimated from these residuals. In a certain sense, this corresponds to the L1 fixed solution in our paper, where we first fix the ambiguities and subsequently estimate the SWE in an additional step. Our study, however, clearly shows that a simultaneous SWE estimation together with ambiguity resolution (L1 SWE fixed) is advantageous especially during wet snow periods.

Regarding page 5, line 23: By applying the assumption of a single water layer there is no need to separate between different snow wetness types at all, which is a great advantage of our approach.
* * *

---

## Author Comment (AC2) · 14 Sep 2018

**Authors response to referee comments**

**An assessment of sub snow GPS for quantification of snow water equivalent**

L. Steiner, M. Meindl, C. Fierz, and A. Geiger

We would like to thank the two reviewers for their constructive feedback and valuable input. Detailed responses are provided below, together with a mark-up manuscript version where the changes made in response to the referee's comments are highlighted.

**Anonymous Referee #1:**

*This study studies snow water equivalent (SWE) from comparing GPS signals between a free receiver and one nearby buried under snow. This study is very well designed, conducted and presented. Very useful results! I have only a few minor suggestions:*

(1) *Figs 1 and 2 show a lot of metal around the measurement site. A few words about (potential) disturbance of the (differential) GPS signals might be useful.*

Indeed, the presence of a lot of metal around the GPS antenna generates multipath effects, disturbing the signal. Multipath is strongest for low elevation signals and is not modelled. The disturbance of the sub snow and reference antenna are not identically and do not cancel out by differential processing. Therefore, multipath effects remain in the GPS observation residuals.

Several sentences are added in section 2.2, 4.1 and 5.4.

(2) *Page 6, line 19: The time shift between the manual and GPS observations is neglected, but might be substantial for warm/melting conditions. Add some evaluation of this effect.*

The time shift is neglected as the SWE is estimated based on a 24 hours GPS data interval, covering the manual time stamp.

A future investigation will show the effect of higher temporal resolution and thus improved time synchronization.

(3) *Page 8, line 2: explain the bold titles and text a bit better. For instance, I don't understand why the second method is called SWE fixed, if SWE is estimated.*

Chosen names of the different strategies are marked in bold, L1 corresponds to the single frequency GPS L1 data, L5 to the Widelane linear combination and fixed corresponds to resolved ambiguities. SWE stands for the simultaneous SWE estimation and ambiguity resolution. So, L1 SWE fixed means: Ambiguities are resolved simultaneously to the SWE estimation based on L1 GPS data.

A paragraph is added to make this point clearer.

*(4) Page 9, line 18 and following. Isn't there an effect to be expected on the SWE estimate whether the snow is dry, or wet (melting). Do you still expect a continuous SWE estimate even for this phase transition? Something in the data about that and discussion of it?*

Yes, we still expect a continuous SWE estimate. The assumption of the single water layer is shown to be adequate for whole seasons including dry and wet snow periods. Differences to the reference sensors occur for both snow types and are due to other not yet modelled error sources (e.g., multipath, flat surface assumption). These effects are investigated in a future study.

Anonymous Referee #2:

*The presented paper shows an interesting and well designed study to highlight the strength and shortcomings of using GPS measurements from sub snow antennas to estimate the snow water equivalent. Nearly three years of GPS data together with reference data are used for this task. This paper brings into application the method described in Steiner et al. (2018, J.Geodesy) where the model of a thin water shell was developed to describe the snow water equivalent. The scope of the study fits the topics of this journal. The paper contains significant new material and interesting results. It is very well structured and thus easy to follow.*
*I have some smaller remarks:*

*(1) The reviewer would like to point the attention of the author to the former IAG study group on site specific effects where among other the impact on snow on the radoms and antennas were studied. Furthermore the following two references could be useful:*

*Jan M. Johansson: Special Study Group 1.158: GPS Antenna and Site Effects.*

*Furthermore the following two references could be useful:*

*S. Vey, A. Güntner, J. Wickert, T. Blume, H. Thoss and M. Ramatschi, "Monitoring Snow Depth by GNSS Reflectometry in Built-up Areas: A Case Study for Wettzell, Germany," in IEEE Journal of Selected Topics in Applied Earth Observations and Remote Sensing, vol. 9, no. 10, pp. 4809-4816, Oct. 2016.doi: 10.1109/JSTARS.2016.2516041*

*S. Tabibi, F. Geremia-Nievinski and T. van Dam, "Statistical Comparison and Combination of GPS, GLONASS, and Multi-GNSS Multipath Reflectometry Applied to Snow Depth Retrieval," in IEEE Transactions on Geoscience and Remote Sensing, vol. 55, no. 7, pp. 3773-3785, July 2017.doi: 10.1109/TGRS.2017.2679899*

Thank you for pointing out the special study group on site specific effects and additional papers. Two references are added.

*(2) Eq.(1) For curiosity: it would be nice to give the typical range of delays.*

If you are interested, typical delay ranges for different snow types can be found at our former paper (Steiner et al., 2018) as well as propagation velocities in these snow types.

*(3) P7.l11 "which takes these correlations into account". Please specify more clearly what is the meaning of sentence and how this is realized when a common estimation is carried out.*

The ambiguities, station height, clocks, and troposphere parameters are expected to be highly correlated to the SWE. In our case, the station height is fixed, the troposphere is

eliminated at the very short baseline, and the clock errors are canceled out by double difference processing. The SWE induced path delay and the ambiguities are remaining. Due to the strong refraction in a water layer model, we expect difficulties in separating them.

The snowpack above the GPS antenna, if not modelled, degrades the ambiguity parameters. The subsequent systematic bias in the float ambiguity parameters leads to false resolved ambiguities and, if fixed for further processing, false estimated SWE. By resolving the ambiguities in the same processing step as estimating the SWE, the software can distinguish these parameters reducing false resolved ambiguities and thus false estimated SWE.

The sentence is reformulated.

*(4) P7.l16 Please could you give a comment on the potential maximum temporal resolution of SWE estimates. Which SWE data rate could reliably be feasible and are there useful applications, such as monitoring intense snow fall during a day or extreme melting?*

The impact of higher temporal resolution will be investigated in a follow-up study. Higher temporal resolution could be beneficious, as the SWE is not stable during snow fall days. A potential application could be to monitor pre season snow fall events, or as you say intense snow fall or extreme melting during a day.

*(5) Table 1. For me, some more explanation how to read the numbers are necessary. Are the values of the first lines (2015/16-2017/18) to combine? If so, please explain how to interpret the number of samples and why b is smaller than all the other values while m is only slightly steeper. Is a weighted average used when combining the "combined" individual years to the overall "combined" solution?*

No weighted average is used when combining the "combined" individual years to the overall "combined" solution.

The values of the first lines (2015/16-2017/18) are computed over the three seasons. So, if you add all number of samples of the separate seasons (204 + 299 + 182 = 685) you get the total number of samples of 685 for all sensors (combined solution).

b smaller, m slightly steeper (2015/16-2017/18):
the slope m of the regression line can be approximately the same, but with a higher offset b. This means, the GPS derived SWE represents the trend of the reference sensors SWE well, but has a higher offset to each of the individual reference sensors. It overestimates the SWE relative to the manual measurements, however, underestimates the SWE relative to the snow scale and snow pillow measurements. Fig.4 illustrates this behavior, where the offsets between the three reference sensors are visible as well, while the general trend of the SWE is followed significantly.

b smaller, m slightly steeper (individual years):
b is much higher for the 2017/18 year compared to the 2015/16 and 2016/17 years, whereas m is approximately the same. Fig. 4 clearly demonstrates this behavior, where the GPS derived SWE has a large offset to the reference sensors by February 2018. The trend, however, follows the reference sensors. This offset is assumed to be caused by an uneven snow surface as it is not present in the previous years.

*(6) Figures 4, 6, 7, 8, 9: If not regulated differently by the journal style file. I would personally prefer increased figure sizes with a larger line size and a larger caption font size for a better readability.*

We agree with you, however, the width of the figures is regulated by the manuscript version. The style will change in the end version.

(7) *P14.l1ff I wonder, the large a posteriori variance factor, especially when comparing to the expected noise of L1 observations of 1-2 mm. My explanation is rather that systematic effects remain in the residuals that yield an increase in the a posteriori variance factor, see also your explanations on P16.l9ff. Do you have any explanation for this problem? We can suppose the antenna positions and the relative distance well known.*

You are right, the expected noise of L1 observations is typically 1-2mm for standard environments. The snow influence on the observations is not fully covered by our model (e.g. flat surface assumption), leading to a higher noise. This behavior is also seen in the residuals, where stochastic and systematic effects remain. Due to these remaining effects, the a posteriori sigma is higher.

(8) *P18. Could you give a typical snow volume, the SWE is representative for?*

The SWE is dependent on the snow density and thus the wetness of the snow volume and is expressed in units of mass per area ($kg/m^2$) or in millimetres of water equivalent (mm w.e.). Consequently, you cannot read the snow volume from a given SWE value, without knowing the snow density.

For example:

Given a new snow (ns), dry snow (ds), or wet snow (ws) sample of $1m^3$ volume:

Typical densities: $\rho_{ns} = 70$ kg/m², $\rho_{ds} = 370$ kg/m², $\rho_{ws} = 700$ kg/m², $\rho_w = 1000$ kg/m²

SWE = h * $\rho_{ns}$ / $\rho_w$

For h = 1m → SWE = $\rho$

SWE$_{ns}$ = 70 mm w.e.

SWE$_{ds}$ = 370 mm w.e.

SWE$_{ws}$ = 700 mm w.e.

(9) *P 21 l22 Please check the reference Rao et al.*

Done.

[revised manuscript text omitted]